# Cancer-Associated Fibroblasts in Undifferentiated Nasopharyngeal Carcinoma: A Putative Role for the EBV-Encoded Oncoprotein, LMP1

**DOI:** 10.3390/pathogens9010008

**Published:** 2019-12-20

**Authors:** Mhairi A. Morris

**Affiliations:** School of Sport, Exercise and Health Sciences, Loughborough University, Loughborough LE11 3TU, UK; M.A.Morris@lboro.ac.uk

**Keywords:** EBV, LMP1, NPC, fibroblast, myofibroblast, cancer-associated fibroblasts, tumour microenvironment, tumour stroma

## Abstract

Undifferentiated nasopharyngeal carcinoma (NPC) is 100% associated with Epstein–Barr virus (EBV) infection, and biopsies display variable levels of expression of the viral oncoprotein, latent membrane protein 1 (LMP1). Emerging evidence suggests an important role for cancer-associated fibroblasts (CAFs) in the NPC tumour microenvironment, yet the interaction between the virus, its latent gene products and the recruitment and activation of CAFs in the NPC tumour stroma remains unclear. This short review will discuss the current evidence for the importance of CAFs in NPC pathogenesis and outline a putative role for the EBV-encoded oncoprotein, LMP1, in governing tumour–stromal interactions.

## 1. Introduction

Undifferentiated nasopharyngeal carcinoma (NPC) is a relatively rare type of head and neck cancer on a global scale, with an incidence of 1.2 cases per 100,000 people, worldwide. However, it exhibits unusually high incidence rates in parts of Southeast Asia, with between 15 and 50 cases per 100,000 in the population [1]. There are numerous causal factors for NPC, including dietary preferences, environmental risk factors and hereditary genetic predispositions, but uniquely, undifferentiated NPC is 100% associated with Epstein–Barr virus (EBV) infection [2]. With recent developments in diagnostic imaging and the use of concurrent chemoradiotherapy, disease control has significantly improved [3]; however, nearly a third of newly diagnosed patients develop local recurrence or distant metastases following treatment [4,5].

EBV encodes nine latent gene products, of which only a handful are expressed in undifferentiated NPC: namely, EBNA1, LMP1, LMP2A, the EBERs and the BARTs [6]. Latent membrane protein 1 (LMP1) is the major transforming protein encoded by EBV and is known for its ability to transform B cells, epithelial cells and fibroblasts in vitro [7] through its constitutive activation of numerous signalling pathways that are frequently deregulated in cancer. These include both the canonical and non-canonical NF-κB pathways [8], the pro-survival PI3K/Akt pathway [9], the stress-related JNK/SAPK pathway [10] and the mitogenic ERK/MAPK [11] and p38-MAPK [12] pathways, which stimulate cell migration and invasion. In addition, LMP1 has been shown to deregulate the pro-fibrotic Smad-independent TGFβ/activin A signalling pathway and overexpress active β1 integrins [13], which may serve to protect epithelial cells from anoikis in order to facilitate metastasis [14].

Controversially, the levels of expression of LMP1 are extremely variable throughout the literature, with reported levels ranging from 20% to 60% [15,16,17]. For years, this has fueled intense debate within the field as to whether LMP1 is indeed implicated in the pathogenesis of NPC. However, findings from Dietz et al. (2004) demonstrated the presence of LMP1 in NPC biopsies that had previously been undetected with conventional IHC by using a tyramid-augmented immunohistochemical (TSA-IHC) technique [18]. In addition, LMP1 was found to induce epithelial hyperplasia in transgenic mice at extremely low levels of expression that are virtually undetectable using current methods. Therefore, it is plausible that even extremely low levels of LMP1 expression may facilitate transformation of NPC cells [19].

NPC is described as a lymphoepithelioma on account of its characteristic inflammatory infiltrate, predominantly made up of EBV-specific cytotoxic T cells and a smaller component of regulatory T cells [20]. However, less is known about other stromal cells that home to the tumour microenvironment (TME) in NPC. Some reports describe a body of activated fibroblasts, also known as cancer-associated fibroblasts (CAFs), surrounding nests of tumour cells in NPC biopsies [21,22], yet the interaction between the virus, its latent gene products and the recruitment and activation of CAFs in NPC pathogenesis is poorly understood. This short opinion piece will review the current body of evidence for the importance of CAFs in the clinical course of NPC, and discuss a putative role for the EBV-encoded oncoprotein, LMP1, in the formation of CAFs in the TME.

## 2. Fibroblasts, Myofibroblasts and Cancer-Associated Fibroblasts

Fibroblasts are one of the most abundant cell types found in the tumour stroma [23,24]. Primarily a source of extracellular matrix (ECM) proteins that provide tissue structure, fibroblasts also serve to determine cell phenotype and function in the wound response to tissue injury [25]. Tumours have been described as “wounds that fail to heal” [26] and fibroblasts have been shown to play an important role in this constitutively active wound response [27]. 

Following tissue injury, fibroblasts become activated to myofibroblasts, secreting cytokines and growth factors, which stimulate basal epithelial cells to proliferate and migrate, and thereby “close the wound”. In addition, they display increased expression and secretion of ECM proteins, which serve as structural support for the migrating epithelial cells, and remodeling enzymes, such as matrix metalloproteinases (MMPs) that help “clear a path” for migration, but myofibroblasts also provide pro-survival signals via outside-in integrin signalling to prevent suspension-induced apoptosis, or anoikis, in migrating epithelial cells [28,29,30]. Numerous different factors have been identified in the fibroblast activation process in both wound healing and tumourigenesis, including TGFβ signalling [31], FGF2 activation [32,33], and β1 integrin expression [34]. Once the wound healing process has finished, activated myofibroblasts are cleared from the site, either by deactivation back to quiescent fibroblasts or by programmed cell death [28,29].

Activated myofibroblasts are characterised by their expression of alpha smooth muscle actin (αSMA)-containing stress fibres and a concomitant lack of smooth muscle markers, such as desmin and smooth muscle myosin, that are commonly expressed by inactive fibroblasts [35]. Myofibroblasts may arise from activated fibroblasts [36], or they can originate from the dedifferentiation of epithelial cells having undergone epithelial-to-mesenchymal transition (EMT) [37]. They can also arise from endothelial-to-mesenchymal transition (EndMT) of local endothelial cells [38,39] or from bone-marrow and tissue-derived mesenchymal stem cells (MSCs) [40].

Tumourigenesis mirrors the wound healing programme in many respects, such as with the secretion and deposition of ECM proteins and recruiting an inflammatory infiltrate replete with immune cells [41,42]. However, the key difference that distinguishes tumourigenesis from the natural wound healing process, which is tightly restricted to a defined area, is the ability of cancer cells to expand or migrate away from the primary site and invade neighbouring tissues.

Cancer-associated fibroblasts (CAFs) also share many similarities to activated myofibroblasts, which have been reviewed extensively elsewhere [24]; however, in brief, CAFs also exhibit increased ECM production, and increased secretion of cytokines and growth factors that promote tumour progression [43,44,45]. CAFs adopt a different immunophenotype when compared with quiescent fibroblasts, which are typically CD34^+^, αSMA^−^ and fibroblast activation protein (FAP)α^−^, whereas CAFs convert to CD34^−^, αSMA^+^ and FAP^+^ [46]. Whilst the majority of CAFs express αSMA, there is some evidence for a small subpopulation of αSMA-negative CAFs in certain cancers, for example pancreatic cancer [47].

CAFs from different cancer types also express high levels of SDF-1/CXCL12, including breast, endometrial and pancreatic cancer [48,49,50]. This can result in elevated levels of MMP2 and MMP9 expression, thereby enhancing cell invasion. Moreover, SDF-1/CXCL12 can also induce cancer cell proliferation via a PI3K/Akt and MAPK signalling-dependent mechanism [49].

## 3. The Role of CAFs in the NPC Tumour Microenvironment

In a timely review of the stromal cellular makeup of the TME of EBV-associated cancers, Tan et al. (2018) briefly mention the presence of CAFs in NPC [51]. However, there is a paucity of research in this particular area, having only recently garnered more interest in the field, despite the anecdotal point noted by Pierre Busson a decade ago in respect of the relative ease with which one can recover stromal fibroblasts when compared with malignant NPC cells cultured in vitro [20]. 

Nonetheless, evidence is emerging in support of the key role that CAFs may play in the pathogenesis of NPC. In a study by Wang et al. (2014), immunohistochemical staining revealed significantly higher levels of αSMA in the stroma surrounding nests of NPC tumour cells, alongside high levels of SDF-1 (also known as CXCL12) in the tumour cells themselves. The authors also demonstrated a correlation between αSMA and CD34 expression—an indicator of microvessel density used as a measure of neoangiogenesis, lending additional support to the role of CAFs in NPC pathogenesis [21]. 

SDF-1/CXCL12 is a secreted factor that is implicated in regulating tumourigenesis and stromal interactions in the TME [52]. In breast cancer models, SDF-1/CXCL12 is known to mediate the growth-promoting effects of stromal fibroblasts on cancer cells [48]. CXCR4 is the cognate receptor for SDF-1/CXCL12 and is also implicated in cancer metastasis [53,54,55]. Indeed, elevated levels of SDF-1/CXCL12 in the TME forms a local concentration gradient along which CXCR4-expressing cells, such as bone marrow-derived mesenchymal stem cells (MSCs) can migrate along, homing to the tumour [56]. 

Malignant NPC cells frequently overexpress CXCR4; however, CXCR4 often localises to the nuclear compartment in malignant NPC biopsies [57], which raises the question whether this may render them unresponsive to CXCL12. Although SDF-1/CXCL12 is known to be released by CAFs in different cancer settings [48,58], it can also be secreted in a paracrine fashion by cancer cells themselves [59], further supporting the findings from Wang et al. (2014). 

In 2017, Chen et al. reported high densities of αSMA-expressing CAFs in less than half of all the NPC biopsies they tested (41.2%). Interestingly, however, they featured in the majority of metastatic NPC tissues (83.3%) and high densities also correlated with shorter overall survival and lower 5-year survival rates, suggesting their utility as independent prognostic factors for NPC survival. Curiously, the authors also describe an unusual occurrence whereby a small number of NPC tumour cells expressed αSMA, which is usually reserved for fibroblasts, smooth muscle cells and perivascular cells [22]. This unexpected observation may suggest that NPC tumour cells undergo conversion via an EMT-like mechanism and adopt an αSMA-expressing mesenchymal phenotype, a phenomenon for which Lee et al. (2013) have already demonstrated a precedent in metastatic lung cancer [60].

In a similar vein, Yu et al. (2018) also demonstrated the use of αSMA expression in CAFs, as well as CD163 expression on tumour-associated macrophages (TAMs), as independent predictors of survival in NPC prognostication using tissue microarrays [61]. To date, this is the only study linking CAFs and TAMs in NPC progression and survival, however studies in other cancers describe a crosstalk between resident CAFs and TAMs in the TME. In one such study, the relationship between pro-inflammatory M2 monocyte TAMs and CAFs in prostate cancer is shown to be reciprocal: the M2 TAMs promote the transdifferentiation of fibroblasts to CAFs, and the CAFs, in turn, facilitate monocyte recruitment and macrophage differentiation [62]. Subsequently, together they promote tumourigenesis and metastasis. Similar synergies are observed in other cancers, including bladder cancer, colorectal cancer, breast cancer and neuroblastoma [63,64,65,66]. The findings presented by Yu et al., whereby the presence of both CAFs and TAMs in NPC correlates with poorer survival rates, suggest that this cooperative crosstalk between the two stromal cell types may occur in NPC as well.

## 4. The Putative Role of LMP1 in Driving CAF Formation in NPC 

Whilst relatively little is known about the mechanistic basis for the role of CAFs in NPC progression, even less is known about the interaction between EBV, its latent gene products and the recruitment and activation of CAFs in NPC. However, there are a number of interesting synergies between the mechanisms involved in CAF-mediated tumourigenesis and the signalling pathways engaged by LMP1. The following section will discuss speculative roles for LMP1 in driving CAF formation in NPC that deserve closer investigation.

Despite the variable levels of LMP1 expression in NPC biopsies reported in the literature, which likely reflect the sensitivity and specificity of the methods of detection used to identify LMP1-positivity [67], it is widely believed that LMP1 expression plays an important role during the early disease process, both in the malignant transformation of nasopharyngeal epithelial cells, as well as recruiting and activating the characteristic inflammatory infiltrate. Unlike late stage disease, where LMP1 expression is highly heterogeneous and often restricted to a small number of NPC cells [68], during early stage disease LMP1 expression is more uniform [69,70], and its expression in premalignant lesions facilitates metastasis earlier in the disease process, suggesting a link to the highly metastatic nature of NPC [71,72].

In an older study, Hu et al. (1995) found LMP1-positive tumours to be more aggressive than their LMP1-negative counterparts, as demonstrated by their faster, more expansive growth and their tendency to invade more frequently at sites outside the nasopharynx. Intriguingly, despite this aggressive phenotype, patients with LMP1-positive tumours had a better prognosis with fewer recurrences [73]. 

In vitro studies have shown that high levels of LMP1 expression are toxic in some epithelial cell lines and that they may induce a host immune response [74,75]. Moreover, even low levels of LMP1 expression have been shown to be sufficient to elicit tumourigenic effects. For example, very low levels of LMP1 expression in the NP69 nasopharyngeal epithelial cell line can induce anchorage-independent growth, morphological alterations and an invasive phenotype in vitro [70], and in transgenic mice, LMP1 has been shown to induce epithelial hyperplasia at extremely low expression levels [19]. In addition to the possibility that even extremely low levels of LMP1 expression can facilitate the transformation of NPC cells, it is plausible that EBV may temporally modulate the expression of LMP1 at different stages of NPC pathogenesis: with elevated levels during early stage disease to facilitate host cell malignant transformation, and subsequently downregulating LMP1 expression in late stage disease to avoid eliciting the host immune response.

In line with previous observations in which NPC biopsies displayed SDF-1/CXCL12-expressing tumour cell nests surrounded by αSMA-expressing CAFs [21,57], it is interesting to note that LMP1 expression in NPC biopsies also correlates with higher levels of CXCR4 and its nuclear localisation, but intriguingly, there was no correlation with SDF-1/CXCL12 expression [76], suggesting that LMP1 does not itself modulate the expression of this growth-promoting chemokine. However, LMP1 has been shown to upregulate CXCR4 mRNA in murine embryonic fibroblasts in an NF-κB-dependent manner [77], and in NPC cells, LMP1 is able to regulate CXCR4 activity through post-translational modification, resulting in enhanced metastatic characteristics [78]. Thus, it is possible that LMP1-mediated overexpression of CXCR4, the cognate receptor for SDF-1/CXCL12, is involved in amplifying the response of NPC cells to CAF-derived SDF-1/CXCL12.

In addition to the well-known signalling pathways engaged by LMP1 that are classically deregulated in cancer [79], more recent studies have identified novel signalling capabilities for this transforming oncoprotein. LMP1 has previously been shown to modulate the expression and secretion of TGFβ1, as well as enhance the TGFβ-dependent modulation of ERK-MAPK [13,80]. TGFβ is a pleiotropic cytokine that displays context-dependent effects on cell growth, differentiation and maturation, and is an important mediator of CAF-driven tumourigenesis [81]. 

In the first instance, TGFβ is known to be involved in activating fibroblasts [31]. During the wound healing process, TGFβ1 induces wound contraction by fibroblasts as well as ECM protein deposition and keratinocyte migration via integrin activation [82]. LMP1 has also been shown to increase the expression and deposition of fibronectin via a Smad-independent TGFβ-mediated mechanism, as well as upregulating the expression of active β1 integrins, the major fibronectin receptor, on the surface of epithelial cells [13]. Therefore, it is intriguing to consider whether LMP1-mediated TGFβ secretion could activate fibroblasts in the NPC TME, enhancing contractility and tumour cell migration.

The secretion of TGFβ1 by CAFs drives EMT in breast cancer cells, resulting in elevated levels of cell-ECM adhesion, breast cancer cell invasion and migration, effects which could be abrogated by inhibiting TGFβ signalling [83]. Similar effects of CAF-derived TGFβ are also seen in other cancers, including gastric, colorectal and bladder cancer [84,85,86,87]. 

By contrast, LMP1 has been shown to induce EMT in epithelial cells via an ERK-MAPK-dependent mechanism, but inhibition of TGFβ signalling had no effect on reversing the EMT phenotype [14]. Taken together, it is tempting to speculate whether LMP1 might drive EMT directly, via ERK-MAPK signalling in epithelial cells, or indirectly via the recruitment and activation of CAFs, stimulating their release of TGFβ to act in a paracrine manner on NPC cells in the TME, or indeed a synergistic combination of the two. These questions remain unanswered at present, but future studies may reveal novel insights into the role of LMP1 in mediating CAF-driven EMT.

Another family of soluble factors that play an important role in CAF-driven tumourigenesis is the fibroblast growth factor (FGF) family of signalling proteins. The secretion of FGF family members by CAFs has been shown in multiple different cancer settings: ovarian CAFs secrete FGF-1, leading to phosphorylation of the receptor, FGFR4 [88]; breast CAFs secrete FGF-2 and signal via both FGFR1 and FGFR2 [89,90]; and colon CAFs secrete FGF-1 and -3, and signal via FGFR4 [91]. CAFs also express a surface-associated FGF2 that is not typically secreted into the extracellular microenvironment. In one study involving the co-culture of colorectal cancer cells with CAFs expressing FGF-2, the phenotype of the cancer cells altered to become more elongated, they were able to migrate longer distances, and invaded into Matrigel supplemented with fibroblasts. This interaction between the cancer cells and the fibroblasts was mediated via αvβ5 integrin [92].

LMP1 has been shown to induce FGFR1 expression and phosphorylation in NP69 nasopharyngeal epithelial cells, as well as upregulating FGF-2, leading to constitutive activation of the FGFR1 signalling pathway. Inhibition of FGFR1 in this cell system abrogated LMP1’s ability to mediate aerobic glycolysis, cellular transformation, cell migration and invasion, thereby suggesting a novel role for FGF2/FGFR1 signalling in NPC pathogenesis. The authors also demonstrated high levels of phosphorylated FGFR1 in NPC biopsies that correlated with LMP1 expression [93]. Therefore, it is interesting to consider whether LMP1-mediated FGF2/FGFR1 signalling is serving as a potential mechanism for amplifying the effect of CAF-mediated tumourigenesis in the NPC TME.

## 5. Conclusions and Future Perspectives 

In summary, CAFs are a prominent feature in NPC biopsies, correlating with poorer prognosis, therefore supporting their role in NPC pathogenesis. NPCs with high levels of LMP1 expression tend to display more aggressive phenotypic features, yet curiously, patients with high LMP1 have a better prognosis. However, since even low levels of LMP1 expression can exert tumourigenic effects, it is possible that this is sufficient to drive NPC transformation and the recruitment and activation of CAFs in the TME. Numerous signalling pathways involved in CAF activation and CAF-driven EMT are also engaged by LMP1, as summarised in Figure 1, yet the link between LMP1 signalling and CAF formation remains unknown.

There are a number of unanswered questions in this area. (1) Could LMP1 be driving the horizontal expansion of NPC tumour cells via transformation of stromal cells in the TME in addition to its already known role in promoting vertical expansion via proliferation and the transfer of genetic and epigenetic changes to progeny cells? (2) How much LMP1 is “enough” to elicit transformation and could even low levels be sufficient to recruit and transform fibroblasts in the TME? (3) Could the NPC cells expressing αSMA have originated from CAFs or even mesenchymal stem cells that have been transformed to adopt a carcinoma-like phenotype? (4) What role, if any, do other EBV-encoded latent gene products play in CAF formation? Therefore, there is a compelling argument for prioritising these research questions in future investigations, which may reveal novel insights into the role of EBV and LMP1 in CAF-driven NPC tumourigenesis.

## Figures and Tables

**Figure 1 pathogens-09-00008-f001:**
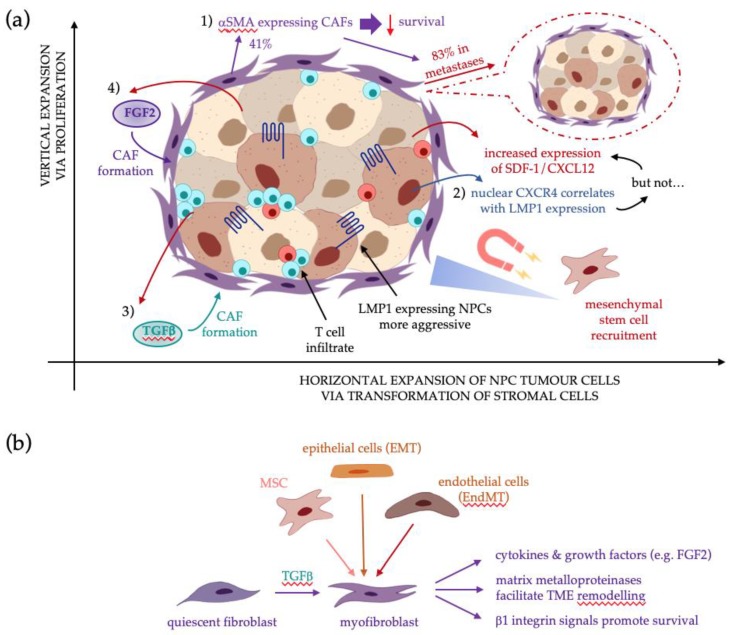
The putative role for LMP1 in CAF formation in NPC: (**a**) From top-left in a clockwise direction: (1) αSMA expression correlates with lower survival rates. High densities of CAFs were seen in 41% of NPC biopsies, yet 83% of metastatic NPC [22]; (2) NPCs display elevated levels of SDF-1/CXCL12 and its cognate receptor, CXCR4. LMP1 expression correlates with nuclear CXCR4 expression, but intriguingly not with SDF-1/CXCL12 [76]. Increased SDF-1/CXCL12 in the TME forms a concentration gradient along which CXCR4-expressing mesenchymal stem cells (MSCs) home to the tumour [56]; (3) epithelial cells expressing LMP1 secrete TGFβ1, a key driver in CAF formation, but its role in NPC CAF genesis remains unknown [13]; (4) LMP1 also modulates expression of FGF2 and its receptor, FGFR1, which suggests it may be implicated in CAF formation [93]; (**b**) Potential sources of CAFs in the TME include quiescent fibroblasts stimulated by TGFβ, mesenchymal stem cells, epithelial cells via EMT, or endothelial cells via EndMT [36,37,38,39,40].

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
