# Peer review of "Cancer-Associated Fibroblasts in Undifferentiated Nasopharyngeal Carcinoma: A Putative Role for the EBV-Encoded Oncoprotein, LMP1"

_pathogens, 2019, doi:10.3390/pathogens9010008_

Round 1

Reviewer 1 Report

This is a concise and interesting review paper describing the role of CAF in the genesis of NPC with relation to LMP1. The authors tried to reveal current understanding of importance of CAFsin the tumor microenvironment using easy to understand English. Overall quality of this review is high enough to attract readers’ attraction. However, it still requires some modifications to improve.

Major comments:

It is not easy to understand and organize topics explained in the manuscript, because the related works are complicated. Authors should draw diagrams to present integrated ideas about the role of LMP1 in promoting CAF formation in NPCs. I requests to remove LMP1-miR146a-CXCR4 story in EBV associated gastric cancers (from line 188 to 194). Usually, cell lines established from gastric cancers show type I latent infection, where some are positive for LMP2A, but none of them shows LMP1 expression (Fields Virology 6th p1946, line 11-14, Proc Natl Acad Sci USA, 91: 9131, 1994, Semin Cancer Biol 7: 175, 1996, Int J Oncol 46:1421, 2015). The GT38 and GT39 cells (Jpn J Cancer Res 89: 262, 1998) used in the Ref. 77 are LMP1 positive cells, showing 43 kDa and 64 kDa of LMP1 protein by immunoblotting. The two bands of full length and truncated form (lytic form) are recognized in LMP1 from B95-8 EBV strain (Virology 251: 273, 1998). As it is reported that CNE1 and HONE1 cells used as NPC cells were contamination of HeLa cells (J Virol 88:10696, 2014), the GT38 and GT39 cells are probably contaminated with B95-8 cells.

Minor comments:

Reference style is not following the instructions by the journal. In some references, authors ask readers to look for another references from cited references. It would be better to reduce such a requirement, because review papers provide important and supporting information from previous studies. More than a third of literatures cited by the author are older than 10 years.

Author Response

Response to Reviewer #1:

Reviewer #1 wrote:

It is not easy to understand and organize topics explained in the manuscript, because the related works are complicated. Authors should draw diagrams to present integrated ideas about the role of LMP1 in promoting CAF formation in NPCs.

My response:

Thank you for the recommendation to enhance the reader’s experience and understanding of the topic by adding diagrams: this has now been addressed with a summarising schematic I created which is included within the conclusion.

Reviewer #1 wrote:

I requests to remove LMP1-miR146a-CXCR4 story in EBV associated gastric canc ers (from line 188 to 194).

My response:

I am in complete agreement (and was unsure about whether or not to include it in the first place!) and therefore have removed this entire paragraph. Thank you for providing such a detailed rationale, which has helped to streamline the manuscript.

Reviewer #1 wrote:

Reference style is not following the instructions by the journal.

My response:

Thank you for picking up on this – my sincere apologies. It appears I had inadvertently selected a check box on Mendeley that caused URLs to be included in every reference. This has been amended and now follows the journal’s conventions.

Reviewer #1 wrote:

In some references, authors ask readers to look for another references from cited references. It would be better to reduce such a requirement, because review papers provide important and supporting information from previous studies.

My response:

I have gone through the manuscript and removed each occurrence of “(reviewed in [X])”.

Reviewer #1 wrote:

More than a third of literatures cited by the author are older than 10 years.

My response:

Whilst I understand this is not ideal, I have tried to cite the most recent relevant research articles, however, this is also a reflection of how little progress has been made in the field of NPC and CAFs in recent years. A number of the very early references are in relation to early studies identifying the signalling capabilities of LMP1, which were conducted in the 1980s and 1990s. I have, however, gone through and added in new references where possible (references 5, 42 and 55 are all new).

Reviewer 2 Report

This review article by Morris explains relationship between CAF and EBV, especially focusing on LMP1, in NPC. The theme is very interesting and text writing is good, too.

Only one possible improvement that the author can do is to prepare a schematic illustration as a figure. Such picture will gather readers' attention and help intuitive understanding of the content. 

Author Response

Response to Reviewer #2:

Reviewer #2 wrote:

The theme is very interesting and text writing is good, too. Only one possible improvement that the author can do is to prepare a schematic illustration as a figure. Such picture will gather readers’ attention and help intuitive understanding of the content.

My response:

Thank you very much for your kind words and for the recommendation to enhance the reader’s experience and intuitive understanding of the content by including a schematic illustration. As noted above, I have created a summarising schematic and included it within the conclusion.